# Biomarkers in Liquid Biopsies for Prediction of Early Liver Metastases in Pancreatic Cancer

**DOI:** 10.3390/cancers14194605

**Published:** 2022-09-22

**Authors:** Anne-Sophie Mehdorn, Timo Gemoll, Hauke Busch, Katharina Kern, Silje Beckinger, Tina Daunke, Christoph Kahlert, Faik G. Uzunoglu, Alexander Hendricks, Florian Buertin, Uwe A. Wittel, Yoshiaki Sunami, Christoph Röcken, Thomas Becker, Susanne Sebens

**Affiliations:** 1Department of General, Visceral, Thoracic, Transplantation and Pediatric Surgery, University Hospital Schleswig-Holstein Campus Kiel, Arnold-Heller-Str. 3, Building C, 24105 Kiel, Germany; 2Section for Translational Surgical Oncology and Biobanking, Department of Surgery, University Medical Center Schleswig-Holstein, University of Lübeck, Campus Lübeck, Ratzeburger Allee 160, 23562 Lübeck, Germany; 3Group Medical Systems Biology, University of Lübeck, Ratzeburger Allee 160, 23538 Lübeck, Germany; 4Institute for Experimental Dermatology, University of Lübeck, Ratzeburger Allee 160, 23562 Lübeck, Germany; 5Institute for Cardiogenetics, University of Lübeck, Building 67, Ratzeburger Allee 160, 23562 Lübeck, Germany; 6Institute for Experimental Cancer Research, University Hospital Schleswig-Holstein Campus Kiel, Kiel University, Arnold-Heller-Str. 3, Building U30 Entrance 1, 24105 Kiel, Germany; 7Department for Visceral, Thoracic and Vascular Surgery, University Hospital Carl Gustav Carus, Technical University Dresden, Building 59, Fetscherstraße 74, 01307 Dresden, Germany; 8General, Visceral and Thoracic Surgery Department and Clinic, University Medical Center, Hamburg-Eppendorf, Martinistraße 52, 20246 Hamburg, Germany; 9Department of General, Thorax, Vascular and Transplant Surgery, University Hospital Rostock, Schillingallee 35, 18057 Rostock, Germany; 10Department of General and Visceral Surgery, Centre of Surgery, Medical Centre, Faculty of Medicine, University of Freiburg, Breisacher Str. 86, 79110 Freiburg, Germany; 11Department of Visceral, Vascular and Endocrine Surgery, Martin-Luther-University Halle-Wittenberg University Medical Center Halle, Ernst-Grube-Straße 40, 06120 Halle, Germany; 12Institute of Pathology, University Hospital Schleswig-Holstein Campus Kiel, Arnold-Heller-Str. 3, Building U33, 24105 Kiel, Germany

**Keywords:** pancreatic ductal adenocarcinoma, PDAC, hepatic metastasis, multiplex analysis, Olink^®^, LEGENDplex^TM^, serum markers, liquid biopsy, inflammation

## Abstract

**Simple Summary:**

Individualized diagnostics approaches in modern cancer therapy require predictive and prognostic biomarkers that are easily accessible and stratify patients for optimal and individualized treatment. Pancreatic ductal adenocarcinoma (PDAC) is still a life-threatening disease mainly because of its late diagnosis in advanced stages or rapid progress even in patients with curative resection of the primary tumor. Moreover, patients with liver metastases exhibit an even worse prognosis. Hence, this retrospective multi-center study aims to identify biomarkers in perioperative serum of PDAC patients predicting early liver metastasis. A highly sensitive biomarker analysis was performed using two different methodological approaches. Olink^®^ analysis, which was also used to validate LEGENDplex^TM^ results, identified significant differences in proteins involved in chemotaxis and migration of immune cells as well as cell growth in serum of patients with early versus late onset of liver metastasis. Further studies with larger cohorts are required to validate these findings for clinical translation.

**Abstract:**

Pancreatic ductal adenocarcinoma (PDAC) is one of the most aggressive solid malignancies with poor survival rates. Only 20% of the patients are eligible for R0-surgical resection, presenting with early relapses, mainly in the liver. PDAC patients with hepatic metastases have a worse outcome compared to patients with metastases at other sites. Early detection of hepatic spread bears the potential to improve patient outcomes. Thus, this study sought for serum-based perioperative biomarkers allowing discrimination of early (EHMS ≤ 12 months) and late hepatic metastatic spread (LHMS > 12 months). Serum samples from 83 resectable PDAC patients were divided into EHMS and LHMS and analyzed for levels of inflammatory mediators by LEGENDplex^TM^, which was validated and extended by Olink^®^ analysis. CA19-9 serum levels served as control. Results were correlated with clinicopathological data. While serum CA19-9 levels were comparable, Olink^®^ analysis confirmed distinct differences between both groups. It revealed significantly elevated levels of factors involved in chemotaxis and migration of immune cells, immune activity, and cell growth in serum of LHMS-patients. Overall, Olink^®^ analysis identified a comprehensive biomarker panel in serum of PDAC patients that could provide the basis for predicting LHMS. However, further studies with larger cohorts are required for its clinical translation.

## 1. Introduction

In Western countries, pancreatic ductal adenocarcinoma (PDAC) is the fourth most frequent cause of cancer-related deaths and represents the most frequent malignancy of the pancreas [1]. With a 5-year overall survival rate of less than 10%, PDAC is one of the most aggressive solid malignancies [1]. Due to a lack of specific symptoms, PDAC is often diagnosed at advanced tumor stages and accompanied by distant metastases. Accordingly, only 20% of the patients are eligible for curative surgical resection of the primary tumor. However, even these patients exhibit a high chance of early recurrence with a visible metastatic burden shortly after or already during the course of adjuvant therapy [2]. Besides the lung and peritoneum, the liver is the leading site of metastasis [3,4]. Moreover, PDAC patients with hepatic metastases (HM) exhibit shorter survival than patients with distant metastases at other sites or local recurrence [4]. To improve the dismal situation of PDAC patients, particularly those with HM, biomarkers that detect the disease at early stages or predict early (hepatic) metastasis are urgently needed.

In this context, liquid biopsies (e.g., from peripheral blood) have gained interest in recent years, as they are non-invasive and easily accessible [5,6,7]. Besides circulating tumor cells (CTC), all kinds of other biological molecules can be used and have been explored as predictive or prognostic biomarkers, also in PDAC, e.g., ctDNA and genetic alterations herein, exosomes, or inflammatory mediators [8].

To date, carbohydrate antigen 19-9 (CA19-9) is the only peripheral blood-based biomarker used for diagnostics of PDAC [9,10,11]. However, as inflammatory conditions such as chronic pancreatitis can also increase serum levels of CA19-9, this marker is neither tumor-specific nor suitable as a diagnostic biomarker for detecting early primary PDAC or early metastatic relapses [12]. However, Izumo et al. recently revealed that serum levels of CA19-9 higher than 37 U/mL prior to tumor surgery are an independent risk factor for early recurrence and poor prognosis of PDAC patients [13]. The recently updated German guideline for PDAC recommends neoadjuvant chemotherapy in PDAC patients with a CA 19-9 value > 529 U/L assuming that a metastatic spread has already occurred [14].

A pilot study with 20 resectable PDAC patients demonstrated that patients with CTC in portal vein blood did not exhibit a different overall or disease-free survival compared to CTC-negative patients. Still, the presence of CTC in portal vein blood was correlated with a higher rate of liver metastases after resection of the primary tumor [15]. In line with this study, analyzing the portal vein blood of sixty patients with periampullary cancer or PDAC revealed that 11/13 patients with a high CTC count in the portal vein blood developed liver metastases within six months after surgical removal of the primary tumor. In contrast, only 6/47 patients with a low CTC count developed liver metastases [16]. Similarly, another study investigating the predictive value of CTC for metastasis in 24 PDAC patients showed that a high count of EpCAM + CD45-cells in the portal vein blood could serve as a predictive biomarker for disease-free survival and occurrence of postoperative metastases [17]. A recent more extensive study comprising 98 PDAC patients underscored the potential of CTC as a predictive biomarker for early metastasis in PDAC by clearly demonstrating that all patients being positive for CTC in peripheral blood developed distant metastases [5]. While CTCs were only detectable in 7/60 patients with metastases, 6/7 CTC-positive patients developed liver metastases indicating a predictive power of CTC, particularly for liver metastasis [5].

For prediction of early metastasis, Guo et al. profiled the ctDNA in preoperative plasma of resectable PDAC patients and were able to demonstrate that mutations in the plasma *KRAS* p.G12D are a suitable predictive biomarker for early distant metastases and associated with shorter survival times [18].

However, since PDAC patients with HM have a particular worse prognosis and only a few studies have been conducted so far exploring biomarkers in peripheral blood of resectable PDAC patients for prediction of (early) liver metastasis, the aim of this study was to identify a biomarker (panel) allowing prediction of early HM. For this purpose, 83 resectable patients with histologically confirmed PDAC were enrolled in this retrospective multi-center study, to analyze peripheral blood by LEGENDplex^TM^ as well as Olink^®^ analysis for the presence of inflammatory mediators.

## 2. Materials and Methods

### 2.1. Study Design and Study Population

Biobanks from seven board-certified German tertiary referral centers for PDAC were screened for eligible patients. Patients diagnosed with PDAC without metastatic spread at the time of diagnosis and who underwent curative surgery (R0 resection) without previous neoadjuvant therapy between October 2007 and December 2019, either by Whipple procedure or left pancreatic resection, were considered eligible for the study. Further inclusion criteria were the development of isolated HM, either early (≤12 months, early hepatic metastatic spread (EHMS)) or late (>12 months, late hepatic metastatic spread (LHMS)) after surgery, and enough serum from the patients. Exclusion criteria were malignancy other than PDAC, primary R1-resection, metastatic spread to different sides than the liver, invalid follow-ups, or inconclusive clinical courses. Clinical courses and oncological data were retrieved from prospectively maintained clinical research databases of local biobanks. A R0 resection was defined as distance to resection margin >0.1 cm, while a R1 resection was present when the resection margin was involved [14].

All patients provided informed written consent. The study was conducted according to the principles of Helsinki. Local Ethics Committees of all participating centers had approved the study (Medical Faculty, University of Kiel and the University Hospital Schleswig-Holstein, Campus Kiel (Reference No. A110/99), Ethics Committee University of Lübeck (16-281), Ethics Committee University of Hamburg (PV3548), Ethics Committee University of Rostock (HV 43/2004 and A 2018-0054, respectively), Ethics Committee University of Halle (2015-106 and 2019-037, respectively), Ethics Committee University of Dresden (EK 76032013), Ethics Committee University of Freiburg (DKRS00014813 and DKRS00007561, respectively)).

### 2.2. Outcome Measures

The primary endpoint was EHMS (≤12 months) or LHMS (>12 months) after R0-resection for PDAC. Secondary endpoints included demographic and oncological data. Further, patients’ serum samples were analyzed for inflammatory markers.

### 2.3. Sample Collection and Analysis

All serum samples analyzed were obtained from patients preoperatively. Although no uniform serum sample quality testing was performed across all biobanks, each biobank providing material for the present study followed established standard operating procedures (SOP) to ensure consistent biospecimen quality.

#### 2.3.1. CA19-9 Analysis

CA19-9 analysis was performed as part of routine diagnostics using the Roche analysis kit according to the manufacturer’s instructions (ElectroChemiLumineszenz ImmunoAssay “ECLIA”, Roche Immunoassay Analyseautomaten COBAS 8000) at the Dept. of Clinical Chemistry, University Hospital Schleswig-Holstein, Campus Kiel, and Campus Lübeck and local Department of Clinical Chemistry at the University Hospitals Hamburg, Rostock, Halle (COBAS e 801) and Freiburg, Germany.

#### 2.3.2. LEGENDplex^TM^ Analysis

The bead-based immunoassay LEGENDplex^TM^ uses the principle of a sandwich immunoassay, in which a soluble analyte is detected by two antibodies. Here, the LEGENDplex^TM^ Human Angiogenesis Panel 1 Mix and Match Subpanel and Human CD8/NK Panel 1 Mix and Match Subpanel (BioLegend, San Diego, CA, USA) were used to determine different human inflammatory mediators (Interleukin (IL)-2, IL-4, IL-6, IL-8, IL-10, IL-17A, Tumor Necrosis Factor-alpha (TNF-α), sFAS, sFASL, Interferon-gamma (IFN-γ), Granzyme A, Granzyme B, Perforin, Granulysin, Vascular Endothelial Growth Factor (VEGF)) in serum of PDAC patients following manufacturer’s instructions. The assay sensitivity or minimum detectable concentration in serum was IL-2: 1.1 pg/mL, IL-4: 0.9 pg/mL, IL-6: 1.0 pg/mL, IL-8: 0.9 ± 0.3 pg/mL, IL-10: 0.7 pg/mL, IL-17A: 1.1 pg/mL, TNFα: 0.6 pg/mL, sFAS: 3.6 pg/mL, sFASL: 1.3 pg/mL, IFNγ: 1.6 pg/mL, Granzyme A: 1.2 pg/mL, Granzyme B: 5.6 pg/mL, Perforin: 2.1 pg/mL, Granulysin: 5.2 pg/mL and VEGF 0.7 ± 0.5 pg/mL. All recombinant proteins were tested individually at the indicated concentrations (VEGF: 5 ng/mL, IL-8: 10 ng/mL, others: 100 ng/mL). No or negligible cross-reactivity was found. Patients’ serum samples were diluted 2-fold with assay buffer and 25 µL of diluted serum samples were used for analysis. The measurement was performed on the BD FACSVerse™ flow cytometer (Beckton Dickinson, East Rutherford, NJ, USA). The evaluation was conducted with the help of the LEGENDplex^TM^-data analysis software v8 (BioLegend, San Diego, CA, USA).

#### 2.3.3. Olink^®^ Analysis

Protein analysis of serum samples was performed using the immuno-oncology 96-plex immunoassay panel (Olink^®^ Proteomics, Sweden). The Olink^®^ technology detects low abundance proteins in plasma and serum and quantifies proteins in wide concentration ranges (Citations). Dual antibody recognition of target proteins results in highly specific immunoassays with a lower limit of quantification < 0.1 pg/mL. Mean intra- and inter-assay variations were reported to be 8.3% and 11.5%, respectively [19]. Briefly, proximity extension assays (PEA) use antibody-pairs labeled with complementary oligonucleotide strands. Two strands create a deoxyribonucleic acid (DNA) sequence which is subsequently extended by DNA polymerase after binding to the same target protein.

In the first step of the PEA assay, 1 µL of serum is diluted with 99 µL of diluent in a 96-well plate. For extension and amplification, 96 µL of extension mix is added, incubated for 5 min at room temperature, and incubated in the polymerase chain reaction (PCR) cycler (Primus 96 plus, MWG Biotech, extension for 20 min at 50 °C, hot start for 5 min at 95 °C, 17PCR Cycles for 30 s at 95 °C, 1 min at 54 °C and 1 min at 60 °C). The PCR products were transferred to a 96 Dynamic Array IFC for Gene Expression (Fluidigm, Corporate Headquarters, South San Francisco, CA, USA) where they were mixed with assay-specific primers in the IFC control Juno™ (Fluidigm, Corporate Headquarters, South San Francisco, CA, USA). In the final step, the amplified DNA reporter sequence for each detected protein is quantified by high-throughput real-time qPCR in the Fluidigm Biomark™ HD System (Fluidigm, Corporate Headquarters, South San Francisco, CA, USA) [19,20,21].

### 2.4. Statistical Analysis

GraphPad Prism version 9.3.1 (350) (GraphPad Software, San Diego, CA, USA) and SPSS 28.0.0.0 (190) (IBM, Armonk, NY, USA) for Mac were used for statistical analysis. Groups of datasets were tested for normal distribution and equal variance by Shapiro–Wilk and Equal Variance tests, respectively.

Normally distributed continuous variables, usually following a Gaussian distribution, are presented as mean ± SD, and groups were compared using the Students’ *t*-test (age, number of lymph nodes, and lymph nodes infiltrated, respectively). Non-normal distribution was assumed in independent continuous variables, not following a Gaussian distribution. They were analyzed using the Mann–Whitney U test (predicted concentration of markers tested by LEGENDplex^TM^ and Olink^®^ analysis). Chi-square test was used for categorical variables (sex, UICC-stage, TNM and LVPn stage and grading, respectively). A logistic regressions model was used to estimate the probability of EHMS or LHMS on one or more predictor variables. The variables included clinical data (age, gender), oncological data (UICC stage, TNM stage, LVPn stage, grading), and inflammatory markers tested by both the LEGENDplex^TM^ and Olink^®^ analysis. A multivariate model building was performed using a stepwise variable selection procedure (inclusion: *p*-value of the score test ≤ 0.05, exclusion: *p*-value of the likelihood ratio test > 0.1). Results are presented as Odd’s ratio (OR) with a 95%-confidence interval (CI) and *p*-value of the likelihood ratio test. Kaplan–Meier-Analysis was used to analyze survival data and further analyzed using log-rank-test [22]. Survival was defined from surgery to death or last contact, whatever occurred first. Tumor recurrence was defined as morphologically proven hepatic tumor recurrence by CT or MRI scan or last contact whatever occurred first.

LEGENDplex^TM^ analysis was performed using BioLegend statistics program and GraphPad Prism.

Quality control and calculation of the normalized protein expression (NPX) for Olink^®^ analysis were performed according to the manufacturer’s recommendations. In detail, four internal controls, which consist of the pooled plasma samples were added to each sample to monitor both the quality of the plate as well as the quality of individual samples. The quality of each sample plate was evaluated based on the standard deviation not exceeding 0.2 NPX of the internal controls. The per sample quality was assessed from the deviation from the median value of the controls for each individual sample. Samples that deviated less than 0.3 NPX from the median passed the quality control.

The NPX was calculated in three steps by the Olink^®^ software. From the delta Ct value, the difference between the analyte and the assay control, the median of the inter-plate controls was deducted, defined as ddCt (analyte). This value was then deducted from an analysis-specific correction factor to obtain the NPX per protein, which is an arbitrary unit on a log2-scale where a high value corresponds to a higher protein expression. Since the specific correction factor is not obtainable for the user the NPX value was used as it was calculated by the software.

Differential protein expression has been calculated using a moderated *t*-test as implemented in the R limma package (Version 3.5) [23]. Limma uses a test similar to the Student’s *t*-test in that it compares the means of NPX values for two groups of replicates for a given protein. The difference is in the calculation of variance. A Student’s *t*-test calculates the variance from the data per protein. The moderated *t*-test uses information from all of the proteins to calculate variance. Differential pathway activity was assessed via a gene set enrichment analysis on the protein fold changes between EHMS and LHMS as implemented in the R gage package (Version 2.44) using the Gene Ontology Biological Processes as gene sets (taken from MSigDB version 7.5) and *p*-value summarization via Stouffer’s method [24,25]. For the analysis only sets being represented by at least 3 proteins were considered. GSEA focuses on the differential expression of sets of related genes. It has advantages over per-gene based different expression analyses, including greater robustness, sensitivity, and biological relevance, as it calculates whether small, yet concerted changes in a set of biological entities, here proteins, show significant regulation as a whole.

## 3. Results

### 3.1. Patients and Demographics

Eighty-three PDAC patients met the inclusion criteria (Figure 1, Appendix A) with the R0 resection status being mandatory. EHMS was present in 52 patients (62.7%) and LHMS in 31 patients (37.3%) (Figure 1, Table 1).

The average age of all patients included was 65.6 ± 10.2 years and patients were equally divided between the two sexes (male 51.8%). This distribution was similar in EHMS and LHMS patients (Table 1). UICC-stage IIB was detected in the majority of patients with pT3 being the main tumor stage (*n* = 53, 64.6%) (Appendix A). Overall, 59.6% of EHMS patients presented with pT3 tumor, while the percentage of LHMS patients with a pT3 tumor was slightly but not significantly higher (73.3%) (Table 1). Overall, no significant differences were found between EHMS and LHMS patients regarding clinicopathological data (Table 1).

In the entire study cohort, overall survival was 14.5 ± 15.1 months and recurrence-free survival was 11.8 ± 9.0 months (Appendix A). However, recurrence-free, death-censored survival and overall survival were significantly poorer in PDAC patients with EHMS compared to LHMS patients (Figure 2A–C).

In order to identify factors predicting EHMS and LHMS, an univariate analysis was performed. The absence of venous infiltration (V0) was the only parameter identified to be predictive of LHMS (Appendix A). Overall, these data indicate that the TNM stage does not help to predict the early appearance of liver metastases in PDAC patients.

### 3.2. LEGENDplex^TM^ Analysis of 14 Inflammatory Mediators

#### 3.2.1. Serum Levels of Inflammatory Mediators Determined by LEGENDplex^TM^ Analysis and Correlation with Early and late Emergence of Liver Metastases

To identify factors correlated with EHMS and LHMS in PDAC patients, serum samples were analyzed for inflammatory mediators as well as the serum biomarker CA19-9 [26]. Values from three patients (EHMS: *n* = 2, LHMS: *n* = 1) were inconclusive and hence excluded from further analysis. Despite a trend towards slightly higher CA19-9 levels in patients with EHMS, no significant difference was detected between both cohorts (Figure 3A). Differences between the two groups for IL-2, IL-4, IL-8, IL-10, IL-17A, TNF-α, Granzyme A, Granzyme B and Granulysin (Appendix A) were insignificant. However, inflammatory mediators known to be involved in tumor progression and angiogenesis, such as IL-6 (Figure 3B) and VEGF (Figure 3C) were elevated in patients with EHMS, which was also observable for IFN-γ (Figure 3D). Interestingly, analysis of sFAS (Figure 3E) and Perforin (Figure 3F) showed significantly higher (*p* = 0.0102 and *p* = 0.0042, respectively) serum levels in patients with LHMS.

#### 3.2.2. Correlation of Serum Biomarkers Identified by LEGENDplex^TM^ Analysis with Clinical Data

In order to identify predictive correlation levels of serum markers and clinical data, uni- and multivariate analyses were performed. Yet, only the absence of venous infiltration (pV0) and elevated perforin levels were predictive for LHMS spread in PDAC patients, in both uni- and multivariate analysis after excluding potential confounders (Appendix A). However, IL-4 and sFASL had to be excluded due to inconclusive values (Appendix A).

### 3.3. Olink^®^ Analysis of 92 Inflammatory Mediators

In the next step, we aimed to validate and extend our results obtained by LEGENDplex^TM^ analysis by performing comprehensive protein biomarker analyses using the immuno-oncology 96-plex immunoassay panel Olink^®^. Olink^®^ analysis comprised the detection of 92 analytes in serum samples from 83 patients.

#### 3.3.1. Quality Control and Correlation of Olink^®^ Analysis and LEGENDPlex^TM^ Analysis

Quality control of the data showed warnings for two samples that were excluded from subsequent analysis. Olink^®^ analysis showed a highly patient-specific protein abundance. Despite normalization, there was a significant trend from low to high mean protein abundance among all patient samples derived from the seven sites (Appendix A), which were also evident in the first principal component of a principal component analysis of the NPX values (Appendix A). Importantly, there was a mostly positive correlation between the common proteins from the LEGENDplex^TM^-array and the Olink^®^-platform. Especially, for IL-6 and IFN-γ strong correlations were identified between the two methods (Appendix A). As sFAS and Perforin were not part of the chosen Olink^®^ analysis panel, these analytes were not used for the comparison. Weak or negative correlation might stem from inaccurate or missing data in the LEGENDPlex^TM^-assay.

#### 3.3.2. Correlation of Serum Biomarkers with the Early and Late Emergence of Liver Metastases

In line with our results shown in Figure 3A, Olink^®^ analysis identified almost similar CA19-9 values in both PDAC patients with EHMS and LHMS, respectively (Figure 4 upper row, Appendix A). Furthermore, Olink^®^ analysis determined slightly higher levels of IL-6, VEGF and IFN-γ in serum of EHMS patients (Figure 4 upper row, Appendix A), confirming these slight tendencies previously detected by LEGENDplex^TM^. Yet, Olink^®^ analysis identified significantly higher serum protein levels of ADA, CASP8, CCL3, CCL20, CD40, CD40LG, FGF2, IL-8, MCP-3, MCP-4, NCR1, PTN and TNFRSF12A, in PDAC patients suffering from LHMS (Figure 4, Appendix A, Appendix A).

#### 3.3.3. Correlation of Serum Biomarkers Identified by Olink^®^ Analysiswith Clinical Data

Next, uni- and multivariate analyses of significantly differential serum markers in PDAC patients with EHMS and LHMS determined by Olink^®^ analysis were performed (Appendix A). CCL3, CCL20, FGF2, IL8, MCP3 and TNFRSF12A were identified as predictive for LHMS in univariate analysis. Yet, after excluding potential confounders, none of these factors could be confirmed by multivariate analysis (Appendix A).

### 3.4. Gene Set Enrichment Analysis

In a final step, a gene set enrichment analysis was performed using the gene ontology biological processes as gene sets. Compared to the late metastasizing group, we found only the GO-terms “cell maturation” as significantly up-regulated in the serum of PDAC patients with EHMS (*p*-value = 0.047). Contrary to this, factors that were elevated in the LHMS group could be associated with diverse processes: besides activation of MAPK activity or cellular responses to abiotic stimuli or estradiol factors could be associated particularly with the migration of lymphocyte and mononuclear cells but also with chemotaxis of natural killer cells (Figure 5). Overall, this analysis reveals a strong association of significantly elevated factors in serum of LHMS patients with processes involved in the recruitment and migration of immune cells.

## 4. Discussion

PDAC still has a dismal prognosis, and even after curative surgical resection, recurrence occurs early and is the most limiting prognostic factor [1,2,3]. However, no reliable prognostic marker for recurrence is currently available, especially for hepatic metastasis, is available. Until today, CA19-9 is the only serum marker used during follow-up of PDAC patients [26,27]. However, CA19-9 is not tumor-specific and is also elevated in patients suffering from pancreatitis or other malignancies and also negative in some patients [11]. Moreover, our study using two different methodological approaches revealed that CA19-9 is also not a suitable marker to differentiate between EHMS and LHMS PDAC patients.

In this multi-center study, we aimed to identify liquid biopsy markers that allow the prediction of EHMS and LHMS in PDAC patients. For this purpose, serum samples from 83 patients were analyzed by two independent technologies, LEGENDplex^TM^ and Olink^®^ analysis. The results were then compared and correlated with clinicopathological data.

LEGENDplex^TM^ analysis identified a trend toward higher serum levels of IL-6, VEGF, and IFN-γ in patients suffering from EHMS, which could be validated by Olink^®^ analysis. Considering the important tumor biological function of either protein, the elevation of these factors is plausible in serum of PDAC patients with an early progression towards HM-burden.

IL-6 is a pleiotropic protein promoting several processes involved in tumor initiation and progression, e.g., proliferation, survival, and invasion of tumor cells [28,29,30]. Accordingly, elevated IL-6 levels have been associated with a less favorable prognosis for different malignancies, including PDAC [29,31]. Furthermore, our data support and extend the study by Kim et al. demonstrating elevated serum levels of IL-6 as an independent risk factor for HM [32]. In this context, the influence of IL-6 on the tumor microenvironment is of pivotal importance, too, as it promotes angiogenesis and impairs anti-tumor immunity [29,33,34].

Likewise, elevated VEGF levels were detected in tumor tissues and in the serum of PDAC patients correlating with poor prognosis [35,36]. VEGF is a well-characterized pro-angiogenic factor involved in tumor angiogenesis and promotes tumor growth and tumor cell dissemination [37]. Additionally, our group could identify VEGF as a factor promoting the reversal of a dormant cell stage and proliferation of PDAC cells in an inflamed hepatic microenvironment, thereby enhancing metastatic outgrowth [38]. Moreover, in an in vivo mouse model, we demonstrated that a laparotomy, which is the surgical procedure used for resection of primary PDAC, leads to an inflammatory response in the liver, involving elevation of several inflammatory mediators, such as TNF-α, IL-1β, IFN-γ and VEGF [39]. These data are consistent with the study by Yang et al. detecting postoperative increases in several inflammatory mediators in the serum of PDAC patients, among others, VEGF [35]. These findings strongly support the view that resection of the primary tumor leads to a systemic inflammatory response involving the liver. This view is further supported by preclinical and clinical studies demonstrating that perioperative interventions aiming at reducing surgery-mediated inflammation might help to minimize recurrence rates [40].

IFN-γ was another factor identified at elevated levels in serum of PDAC patients with EHMS using LEGENDplex^TM^ analysis. IFN-γ is an essential cytokine in innate and adaptive immunity. It activates macrophages and represents a potent cytotoxic effector molecule released by natural killer (NK) cells and effector CD8+ T-cells [41,42]. Given the critical immune-stimulatory role of this cytokine in immunosurveillance of cancers [42], its elevation in serum of PDAC patients with early progression seems at first sight inconclusive. However, it is also known that IFN-γ triggers chemotherapy resistance in PDAC patients [43], promoting survival and further expansion of tumor cells.

LEGENDplex^TM^ analysis also revealed significantly elevated levels of sFAS and Perforin in the serum of PDAC patients with LHMS. The FAS receptor exists as a membrane-bound and soluble form (sFAS). While the former leads to apoptosis induction upon binding of FAS ligand, sFAS is thought to inhibit FAS ligand-mediated apoptosis induction. Elevated serum sFAS levels have been demonstrated in various cancers and correlated with disease progression and poor prognosis [44,45]. Our findings demonstrating higher serum sFAS concentrations in PDAC patients with LHMS compared to EHMS patients might indicate another role of sFAS in PDAC. Whether this marker is a reliable predictor of LHMS in PDAC patients requires further validation.

Finally, LEGENDplex^TM^ analysis revealed significantly elevated levels of Perforin in the serum of LHMS patients. This finding aligns with the critical role of perforin as one of the main cytotoxic effector molecules released by NK and effector CD8+ T cells. It is also a central factor in immune surveillance by which the host prevents metastatic outgrowth. Accordingly, the later emergence of visible liver metastases can be regarded as a longer-lasting control by the immune system (involving Perforin mediated elimination of certain tumor clones) to control the growth of disseminated PDAC cells in the liver. In line with this view, the gene set enrichment analysis revealed that factors which were elevated in the serum of LHMS patients could be associated with immune processes involved in the recruitment and migration of immune cells which contributes restraining metastatic outgrowth.

Although we did not validate the findings of sFAS and Perforin by Olink^®^ because these two proteins were not included within the immune-oncology marker panel, Olink^®^ analysis confirmed most of the results obtained by LEGENDplex^TM^. Especially regarding IL-6, VEGF and IFN-γ, Olink^®^ confirmed the previous trends identified by LEGENDplex^TM^. However, some discrepancies could also be noted, e.g., for IL-8, which was detected at significantly elevated levels in LHMS patients by Olink^®^ but not by LEGENDplex^TM^. The technology-based differences in the detection of markers in serum of both patient cohorts can be explained by the test principle and associated with the different sensitivity and specificity of the two technologies. LEGENDplex^TM^ is a sandwich immunoassay in which specific analytes bind to so-called capture beads and become detectable after binding to a fluorescent dye. Detection is performed using a standard flow cytometer. The main limitations of this technology are the comparably large amount of patient material needed (12.5 to 25 µL/patient), the inter-array variability, and the accumulation of cell clusters, which may lead to misleading detections of signals and misinterpretation of the data.

Olink^®^ proximity extension assay (PEA), on the other hand, is a high throughput, multiplex immunoassay that is reliably working by using of only 1 µL of sample volume. For each protein marker a specific pair of antibodies binds to specific binding sites of the respective protein. The antibodies are, in turn, coupled to unique oligonucleotides (DNA probes) that come into contact with each other (hybridization) when two matching antibodies have bound to a protein. The hybridized oligonucleotides are amplified using DNA polymerases and detected using qPCR. During the final step, each hybridized oligonucleotide is detected individually for every sample (9.216 individual reactions). Furthermore, Olink^®^ implements multiple internal and external controls to ensure reliable results. By this, Olink^®^ reaches a high specificity as well as sensitivity and overcomes the limitations of a standard multiplex immunoassay such as the LEGENDplex^TM^.

Accordingly, extending the analysis of the same serum samples via Olink^®^ analysis, a further panel of markers could be determined to be significantly higher in serum of LHMS patients, namely ADA, CASP8, CCL3, CCL20, CD40, CD40LG, FGF2, IL-8, MCP-3, MCP-4, NCR1, PTN, and TNFRSF12A. Elevated ADA levels have already been described in the serum of PDAC patients correlating with tumor size and metastases. Moreover, ADA serum levels increased in healthy controls to patients with acute and chronic pancreatitis and finally in PDAC patients indicating that ADA may play a role in inflammatory processes of the pancreas [46]. Interestingly, in our study, ADA levels were higher in PDAC patients with LHMS compared to EHMS patients indicating another role of ADA in the liver. Indeed, an essential role for ADA and Adenosine deaminase acting on RNA 1 (ADAR1) has been described in the maintenance of liver homeostasis [47,48]. Accordingly, it can be hypothesized that elevated serum levels of ADA indicate a homeostatic hepatic microenvironment that can control PDAC cell growth thereby preventing the outgrowth of liver metastases [38].

As revealed by our gene set enrichment analysis, many factors being elevated in the serum of LHMS patients, e.g., the chemokines CCL3, CCL20, MCP-3, and MCP-4, are involved in chemotaxis, migration, and recruitment of monocytic and lymphocytic immune cells into the tumor [49,50]. Thus, higher levels of those mediators in LHMS patients may indicate a better immune control of the tumor burden in the liver, providing another mechanism by which outgrowth of visible metastases might be prevented in these patients. This view is further supported by the findings of higher levels of Perforin (see above) and CD40 and CD40LG in the serum of LHMS patients. Furthermore, higher serum levels of IL-8 are also in line with better host-mediated control of PDAC cell growth because IL-8 was identified to promote a dormant stage in PDAC cells, thereby keeping PDAC cells under growth control in a physiological (non-inflamed) liver microenvironment [38]. Accordingly, these experimental data align with elevated IL-8 serum levels in PDAC patients with LHMS and with the study by Yang et al., which identified high serum levels of IL-8 to be associated with a better prognosis [35]. Of note, IL-8 and PTN have been described to be involved in angiogenesis, thus not supporting a predictive role of these factors for LHMS [50,51,52]. The same applies to CCL20, which has been described to induce Epithelial-Mesenchymal-Transition (EMT) in PDAC cells, thereby promoting tumor cell migration and invasion and finally leading to metastasis [53]. However, despite their putative tumor-promoting functions, an increase in these factors appears to be associated with a tumor-suppressive effect and the later occurrence of liver metastases in PDAC patients.

In summary, our study demonstrated significant differences in the presence of inflammatory mediators in the serum of PDAC patients with LHMS compared to EHMS. In order to further increase the reliability and value of the marker panel, further biomarker analyses (either using another Olink^®^ marker panel or another method) are required in order to identify also biomarkers that are elevated in the serum of EHMS patients. For clinical translation it will be optimal to have a specified panel of markers elevated in either EHMS or LHMS. Furthermore, although this multi-center study included samples from seven board-certified tertiary referral centers, the cohort sizes were still small due to the strict inclusion criteria, i.e., R0-resectable PDAC patients with solely HM.

Overall, Olink^®^ multiplex analysis seems to represent a superior, reliable method for comprehensive protein biomarker analyses in the serum of cancer patients. Based on our results, this technology allows a broad and highly sensitive serum biomarker analysis from PDAC patients. The present data suggest its suitability to identify biomarkers that provide information on whether PDAC patients will experience early recurrence with liver metastases or remain stable for longer without liver involvement and thus have a better prognosis. However, for its clinical translation validation studies are needed involving the analysis of the differentially regulated Olink^®^ marker panels in larger cohorts. Further validation will also include analysis of the specified marker panel in longitudinal samples after resection of the primary tumor.

## 5. Conclusions

PDAC is still a life-threatening disease mainly because of its diagnosis at predominantly advanced stages or rapid progress, even in those patients who could undergo curative R0 resection of the primary tumor [2,27]. Especially patients with HM exhibit a significantly worse prognosis than those with metastatic manifestation in other organ sites [2]. Therefore, it is particularly important to identify patients with potentially worse prognosis, such as those patients with a higher risk of HM but also borderline resectable patients in order to provide an optimal therapy, e.g., neoadjuvant treatment. For this purpose, predictive and prognostic biomarkers that are easily accessible in liquid biopsies are of particular interest to better stratify those patients for respective (yet to be defined) therapeutic approaches.

Our study detected and compared biomarker levels in serum from PDAC patients with EHMS and LHMS, respectively, by two independent methods. Using Olink^®^ multiplex analysis, we identified a panel of significantly elevated biomarkers in PDAC patients with LHMS. Since these biomarkers are easily detectable in small amounts of peripheral blood which is routinely taken for diagnostic purposes, this method bears a great potential to be integrated in clinical routines. Yet, these results need to be validated in a larger cohort, potentially replenished with other serum markers, to be thoroughly used as individualized, patient-oriented screening panels.

## Figures and Tables

**Figure 1 cancers-14-04605-f001:**
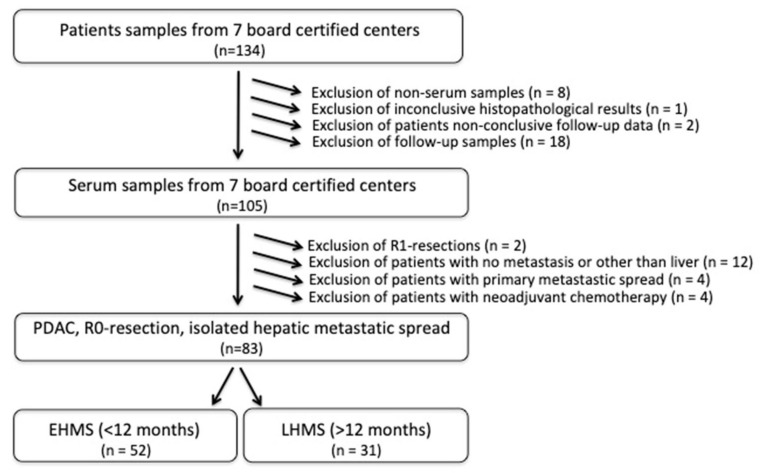
Flow chart of inclusion criteria for PDAC patients with early hepatic metastatic spread (EHMS) and late hepatic metastatic spread (LHMS). PDAC: pancreatic ductal adenocarcinoma.

**Figure 2 cancers-14-04605-f002:**
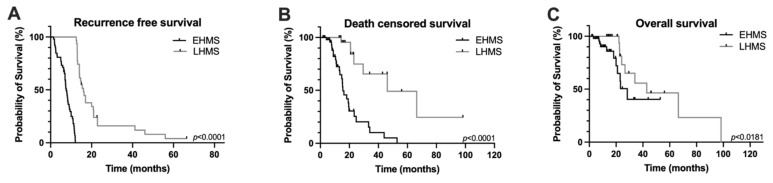
Kaplan–Meier survival curves of PDAC patients included in the study. Recurrence-free survival (**A**), death censored survival (**B**) and overall survival (**C**) were significantly poorer in PDAC patients with early hepatic metastatic spread (EHMS) (*n* = 52) compared to patients with late hepatic metastatic spread (LHMS) (*n* = 31). Statistically significant *p*-values are indicated in the graphs.

**Figure 3 cancers-14-04605-f003:**
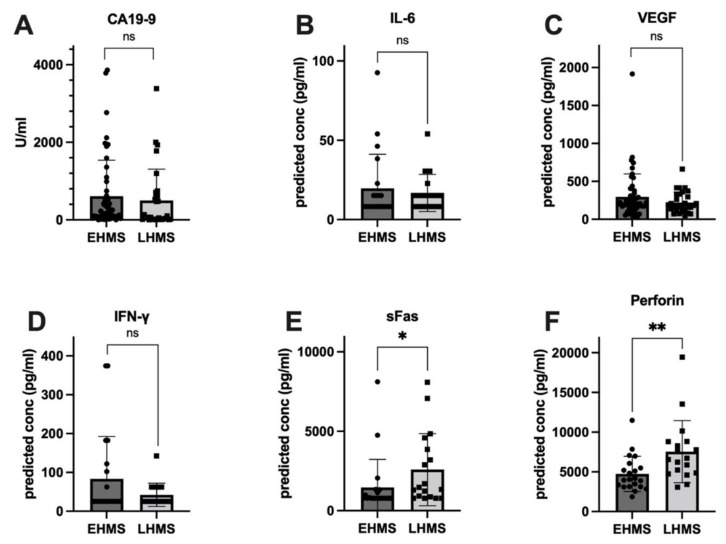
Serum levels of inflammatory markers showing a higher tendency in PDAC patients with early hepatic metastatic spread (EHMS, *n* = 50 and 21, respectively) CA19-9 (**A**), IL-6 (**B**) and VEGF (**C**) or in PDAC patients with late hepatic metastatic spread (LHMS, *n* = 30 and 21, respectively) IFN-γ (**D**), sFAS (**E**) and Perforin (**F**) revealed by Roche Assay (CA 19-9) and LEGENDplex^TM^ analysis (all other factors). Data are presented as median. Statistical analysis was performed using Mann–Whitney U test for non-normally distributed data. * *p*-value < 0.05, ** *p*-value < 0.005.

**Figure 4 cancers-14-04605-f004:**
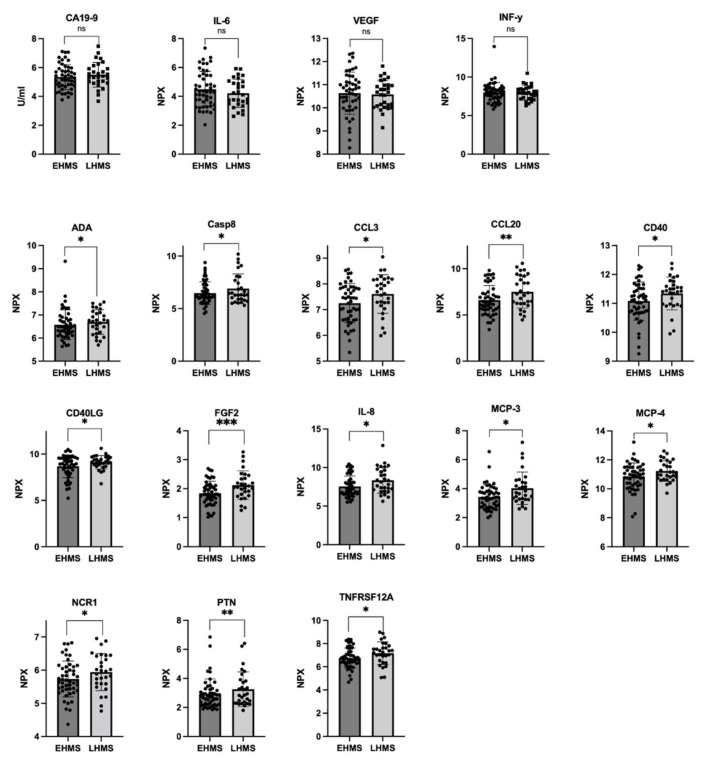
Selected markers determined by Olink^®^ analysis in serum from PDAC patients with early hepatic metastatic spread (EHMS, *n* = 49) or late hepatic metastatic spread (LHMS, *n* = 29). CA19-9 (previously also analyzed by Roche Assay), IL-6, VEGF, and IFN-γ (previously also analyzed by LEGENDplex^TM^) were analyzed by Olink^®^. Using Olink^®^, the following other parameters were identified at different levels in the two cohorts. ADA: Adenosine Deaminase; CA19-9: Carbohydrate antigen 19-9; CASP8: Caspase 8; CCL3: chemokine ligand 3; CCL20: chemokine ligand 20; CD4: cluster of differentiation 4; CD40LG: cluster of differentiation 40 ligand; FGF2: Fibroblast Growth Factor 2; IL-6: Interleukin-6; IL-8: Interleukin-8; IFN-γ: Interferon-γ; MCP-3: Monocyte Chemoattractant Protein-3; MCP-4: Monocyte Chemoattractant Protein-4; NCR1: Natural Cytotoxicity triggering Receptor 1; PTN: Pleiotrophin; TNFRSF12A: Tumor Necrosis Factor Receptor Superfamily Member 12A; VEGF: Vascular Endothelial Growth Factor. Statistical analysis was performed using Mann–Whitney U test for non-normally distributed data. * *p*-value < 0.05, ** *p*-value < 0.005, *** *p*-value < 0.001.

**Figure 5 cancers-14-04605-f005:**
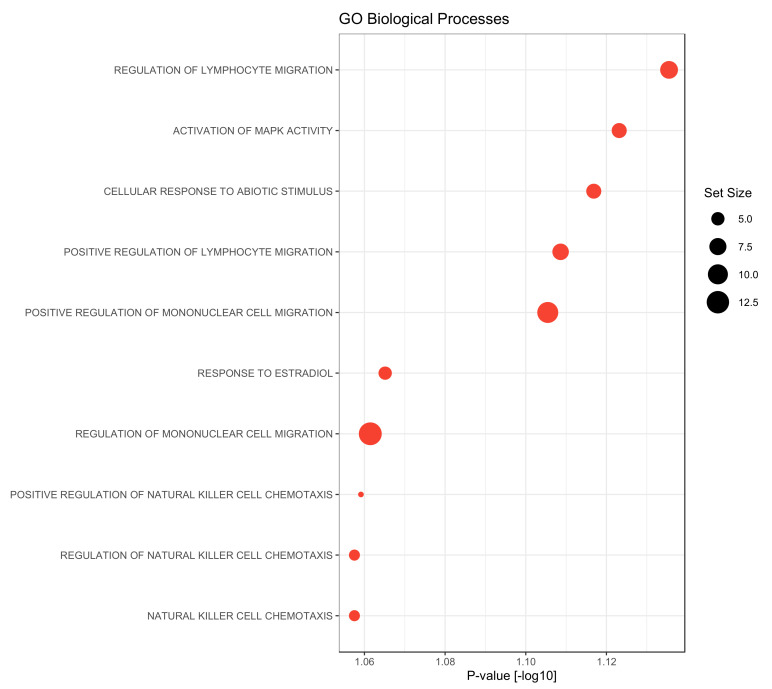
Gene Set Enrichment analysis on biological processes gene sets of the gene ontology (GO) of proteins up-regulated in PDAC patients with late hepatic metastatic spread (LHMS) compared to early hepatic metastatic spread (EHMS). Gene set enrichment was performed using R-library gage (version 2.44) based on the fold change differences between the LHMS and EHMS groups. Gene sets containing at least 3 proteins were considered, only. Dot diameters correspond to the gene set size.

**Table 1 cancers-14-04605-t001:** Demographic and pathological data of PDAC patients included stratified by early (≤ 12 months, *n* = 52) and late (>12 months, *n* = 31) metastatic hepatic spread (EHMS and LHMS). Data are presented as mean ± SD, total numbers (*n*) or relative frequencies (%). Continuous variables were tested using ^a^ Students’ *t*-test (normally distributed), while categorical variables were compared using ^b^ Chi-Square. EHMS: early hepatic metastatic spread; G: Grade; L: Lymphatic infiltration; LHMS: late hepatic metastatic spread; N: Nodal infiltration; Pn: Perineural infiltration; SD: Standard deviation; T: Tumor; UICC: Union international contre le cancer; V: Venous infiltration. ^+^ missing data in some patients (UICC: *n* = 1, T: *n* = 1; N: *n* = 1, L: *n* = 55, V: *n* = 53, Pn: *n* = 28, G: *n* = 10, respectively. missing patient data in some cases).

		EHMS (*n* = 52)	LHMS (*n* = 31)	*p*-Value
Patient demographics				
Age in years (mean ± SD)		65.5 ± 10.9	65.9 ± 9.1	0.840 ^a^
Sex, *n* = males (%)		28 (53.8)	15 (48.4)	0.630 ^b^
Pathological Data *n* (%)				
UICC-stage ^+^				
	IA	1 (1.9)	1 (3.3)	0.690 ^b^
	IB	6 (11.5)	2 (6.7)	0.474 ^b^
	IIA	12 (23.1)	6 (20.0)	0.746 ^b^
	IIB	28 (53.8)	19 (63.3)	0.403 ^b^
	III	5 (9.6)	2 (6.6)	0.645 ^b^
pT	pT1	1 (1.9)	2 (6.7)	0.270 ^b^
	pT2	19 (36.5)	5 (16,7)	0.148 ^b^
	pT3	31 (59.6)	22 (73.3)	0.211 ^b^
	pT4	1 (1.9)	1 (3.3)	0.690 ^b^
N	pN0	19 (35.6)	10 (33.3)	0.770 ^b^
	pN1	19 (36.5)	17 (56.7)	0.077 ^b^
	pN2	14 (26.9)	3 (10.0)	0.069 ^b^
	N+ (%)	62.7	65.6 0.790 ^b^	0.790 ^b^
L^+^	L0	17 (41.5)	9 (52.9)	0.424 ^b^
	L1	24 (58.5)	8 (47.1)	0.100 ^b^
V	V0	32 (78.0)	14 (82.4)	0.713 ^b^
	V1	9 (22.0)	3 (17.6)	0.713 ^b^
Pn	Pn0	6 (2.1)	2 (12.5)	0.783 ^b^
	Pn1	33 (84.6)	14 (87.5)	0.783 ^b^
G	G1	1 (2.1)	0 (0.0)	0.433 ^b^
	G2	21 (44.7)	12 (46.2)	0.729 ^b^
	G3	25 (53.2)	14 (53.8)	0.985 ^b^

## Data Availability

The data presented in this study are available in this article.

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
