# Peer review of "Biomarkers in Liquid Biopsies for Prediction of Early Liver Metastases in Pancreatic Cancer"

_cancers, 2022, doi:10.3390/cancers14194605_

Round 1

Reviewer 1 Report

The authors present a method for the early detection of hepatic spread in pancreatic cancer. 

The research is clearly presented however I have soem questions.

1. How is the quality of the analysed material guaranteed? Any tests performed within the 7 centers for quality of representative material?

2. What is the sensitivity and specificity of the described method?

3. Would it be necessary to continue with the complete Olink biomarker panel in future or is there a selected set of biomarkers? Adding other serum markers remains necessary... this make the method quite complex.

Author Response

We thank the reviewer for the fast review and valuable comments. Accordingly, we have now prepared a revised version of the manuscript. All changes have been highlighted by underlining.

Below, we will explain point-by-point how the arguments and criticisms of the reviewer have been dealt with.

Request 1: “How is the quality of the analysed material guaranteed? Any tests performed within the 7 centers for quality of representative material?”

Answer 1: Although no uniform serum sample quality testing is performed across all biobanks, each biobank providing material for the present study follows established standard operating procedures (SOPs) to ensure consistent biospecimen quality. We have included this sentence in chapter 2.3 “Sample collection and analysis”.

Request 2: “What is the sensitivity and specificity of the described method?”

Answer 2: In the revised version of the manuscript we have provided this information in chapter 2.3.2 and 2.3.3.

Request 3: “Would it be necessary to continue with the complete Olink biomarker panel in future or is there a selected set of biomarkers? Adding other serum markers remains necessary... this make the method quite complex.”

Answer 3: We thank the reviewer for this important comment. In order to address this question, we have modified the last two chapters of the discussion as follows: “In summary, our study demonstrated significant differences in the presence of inflammatory mediators in the serum of PDAC patients with LHMS compared to EHMS. In order to further increase the reliability and value of the marker panel, further biomarker analyses (either using another Olink® marker panel or another method) are required to identify also increased biomarkers in the serum of EHMS patients. For clinical translation it will be optimal to have a specified panel of markers elevated in either EHMS or LHMS.” and “However, for its clinical translation validation studies are needed involving the analysis of the differentially regulated Olink® marker panels in larger cohorts. Further validation will also include analysis of the specified marker panel in longitudinal samples after resection of the primary tumor.”

We very much hope that these modifications and additions sufficiently improved our paper making it now suitable for publication in Cancers.

Thank you very much for your consideration and efforts!

Sincerely yours,

S. Sebens, PhD

Reviewer 2 Report

Pancreatic cancers often first spread within the abdomen and to the liver. They can also spread to the lungs, bone, brain, and other organs later.CA19-9 is the only one to diagnose biomarkers for pancreatic cancer, however it still has some disadvantages such as the non-specific etc. In this manuscript, the authors employed multiple patient specimens that comply with their criteria to do the prognosis analysis between the early and late metastases in the liver. They found that late metastasis status of the pancreatic cancer predicts the bad clinical outcome and through the comparison, they identified a series of factors including some inflammatory cytokines that were highly evaluated in the late metastasis.

Some concerns about the manuscript.

1. The LHMS cohort predicts good survival compared with the EHMS, I am not sure the authors are focusing on the discovery of biomarkers in EHMS or LHMS. In the abstract they identified a comprehensive biomarker panel in serum of PDAC patients that could provide the basis for predicting LHMS. But the evidence in this manuscript that the biomarkers in EHMS are more important.

2. LEGENDplexTM analysis were performed for the biomarkers exploration, and sFAS and Perforin are significantly upregulated in the LHMS, but you did not verify them in the following Olink analysis.Also you conclusion are based on the results of Olink, is it necessary of the LEGENDplexTM part for this manuscript?

3.For the fig5 you also could see the down-regulated in PDAC patients with late hepatic metastatic spread (LHMS) compared to early hepatic metastatic spread (EHMS) which could see some different gene signaling pathway enriched in the EHMS.

4. Heatmap could be used for the differential cytokines expression which could more clearly show your data.

5.Minor errors should be carefully checked, e.g. line 382 two full stops after analysis. Likewise or lieweise in line 939.

Author Response

We thank the reviewer for the fast review and valuable comments. Accordingly, we have now prepared a revised version of the manuscript. All changes have been highlighted by underlining.

Below, we will explain point-by-point how the arguments and criticisms of the reviewer have been dealt with.

Request 1: “The LHMS cohort predicts good survival compared with the EHMS, I am not sure the authors are focusing on the discovery of biomarkers in EHMS or LHMS. In the abstract they identified a comprehensive biomarker panel in serum of PDAC patients that could provide the basis for predicting LHMS. But the evidence in this manuscript that the biomarkers in EHMS are more important.”

Answer 1: We fully agree with the reviewer that this is an important point. We have addressed this issue in the following sentence in the discussion: “In summary, our study demonstrated significant differences in the presence of inflammatory mediators in the serum of PDAC patients with LHMS compared to EHMS. In order to further increase the reliability and value of the marker panel, further biomarker analysis (either using another Olink® marker panel or another method) are required in order to identify also biomarkers that are elevated in the serum of EHMS patients. For clinical translation it will be optimal to have a specified panel of markers elevated in either EHMS or LHMS.”

Request 2: “LEGENDplexTM analysis were performed for the biomarkers exploration, and sFAS and Perforin are significantly upregulated in the LHMS, but you did not verify them in the following Olink analysis.Also you conclusion are based on the results of Olink, is it necessary of the LEGENDplexTM part for this manuscript?”

Answer 2: We agree with the reviewer that this point needs further clarification. For this purpose, we have included the following sentence in the discussion: “Although we did not validate the findings of sFAS and Perforin by Olink® because these two proteins were not included within the immune-oncology marker panel, Olink® analysis validated most of the results obtained by LEGENDplexTM.”

Request 3: “For the fig5 you also could see the down-regulated in PDAC patients with late hepatic metastatic spread (LHMS) compared to early hepatic metastatic spread (EHMS) which could see some different gene signaling pathway enriched in the EHMS.”

Answer 3: We thank the reviewer for this suggestion. There was only one gene set significantly upregulated in EHMS group compared to LHMS, namely cell maturation (p-value = 0.047). We added this information in the results section 3.4 when discussing the GSEA analysis: “Compared to the late metastasizing group, we found only the GO-terms “cell maturation” as significantly up-regulated in the serum of PDAC patients with EHMS (p-value =0.047).

Request 4: “Heatmap could be used for the differential cytokines expression which could more clearly show your data.”

Answer 4: As requested we also show the differentially regulated biomarkers determined by Olink® as Heatmap in the new Supplementary Figure 6.

Request 5: “Minor errors should be carefully checked, e.g. line 382 two full stops after analysis. Likewise or lieweise in line 939.”

Answer 5: As requested we have removed one of the full stops in each line.

We very much hope that these modifications and additions sufficiently improved our paper making it now suitable for publication in Cancers.

Thank you very much for your consideration and efforts!

Sincerely yours,

S. Sebens, PhD

Reviewer 3 Report

The manuscript titled “Biomarkers in liquid biopsies for prediction of early liver metastases in pancreatic cancer” describes the authors wanted to explore effective biomarkers from serum to early predict metastasis of pancreatic cancer. Honestly, the manuscript is not easy to track even though there are only a few analyses involved in the study. I’d suggest the authors emphasize the biomarkers that most like to predict metastasis based on your results. The followings are some concerns and comments have been pointed out that the authors may want to consider.

1.  Line 67 Keywords: None of the listed keywords as below, “multiplex analysis”, “liquid biopsy (only one time of liquid biopsies on line 84)”, “metastatic transition”, appears in the main content. Please consider some more suitable ones if the authors don’t mind.

2.      Lines 89-90: I’d suggest the authors add references to “To date, CA19-9 is the only…biomarker…PDAC”, even if it shares the same reference of [9] or some more others to make it clearer.

3.  Lines 134-135: Please include “12 months” into either the “early” or “late” group.

4.  Line 157: I’d suggest the authors include a brief protocol for CA19-9 analysis.

5.      Line 159: The left half “quote” is upside-down. How did you do that?

6.  Line 163: I’d suggest the authors include at least a brief method for LEGENDplexTM analysis.

7.  Line 205: Please use italic p as it refers to a p-value throughout the manuscript.

8.      Line 244: I’d suggest the authors use “years old” for age instead of “years” throughout the manuscript. For example, in Table 1, etc.

9.  Line 252 Table 1: I’d highly suggest the authors bracket the percentage to make your data presentation clearer.

10.  Line 293 Figure 3, line 334 Figure 4: Please include a brief statistical description.

11.  Line 361 Figure 5: a) Please include the Figure 5 method in the methods section and make it clearer. b) Please include a clearer description of Figure 5 results and discussion.

12.  Table 1, Supplementary Table 2, Table 4, and other related parts: Did the authors mean clinical data only refers to “age at surgery and gender”? I’d highly suggest the authors consider rephrasing it to make it clearer.

13.  In the current manuscript, LEGENDplexTM analysis (line 163 section 2.3.2) tests the protein level in the serum, and protein level biomarkers; the PCR-based Olink® analysis (line 175 section 2.3.3) shows gene expression level and transcriptional level biomarkers. The gene expression could not exactly predict its protein level. They are under different regulation mechanisms. It’s difficult to see what the advantages are that the authors listed two analyses high throughput versus other high throughput gene or protein analyses for clinical practice. 

Author Response

We thank the reviewer for the fast review and valuable comments. Accordingly, we have now prepared a revised version of the manuscript. All changes have been highlighted by underlining.

Below, we will explain point-by-point how the arguments and criticisms of the reviewer have been dealt with.

Request 1: “Line 67 Keywords: None of the listed keywords as below, “multiplex analysis”, “liquid biopsy (only one time of liquid biopsies on line 84)”, “metastatic transition”, appears in the main content. Please consider some more suitable ones if the authors don’t mind.”

Answer 1: As suggested we have removed “metastatic transition”. As blood is an important compartment of liquid biopsies we wish further remain this keyword. However, we have included the keyword more often in the manuscript text. The same applies to “multiplex analysis”.

Request 2: “Lines 89-90: I’d suggest the authors add references to “To date, CA19-9 is the only…biomarker…PDAC”, even if it shares the same reference of [9] or some more others to make it clearer.”

Answer 2: As requested, we have incorporated two more references.

Request 3: “Lines 134-135: Please include “12 months” into either the “early” or “late” group.”

Answer 3: As requested, we have added “12 months” in the text.

Request 4: “Line 157: I’d suggest the authors include a brief protocol for CA19-9 analysis.”

Answer 4: Since CA19-9 analysis was performed as part of routine diagnostics according to the manufacturer's instructions, we have omitted a detailed description of the protocol but have added the following sentence: “CA19-9 analysis was performed as part of routine diagnostics using the Roche analysis kit according to the manufacturer´s instructions.”

Request 5: “Line 159: The left half “quote” is upside-down. How did you do that?”

Answer 5: Since inn our manuscript version the presentation is correct, we regret that we cannot solve this issue.

Request 6: “Line 163: I’d suggest the authors include at least a brief method for LEGENDplexTM analysis.”

Request 6: We fully agree with the reviewer and have included a brief information on the LEGENDplexTM analysis in chapter 2.3.2.

Request 7: “Line 205: Please use italic p as it refers to a p-value throughout the manuscript.”

Answer 7: As requested, we use only italic p when it refers to a p-value.

Request 8: “Line 244: I’d suggest the authors use “years old” for age instead of “years” throughout the manuscript. For example, in Table 1, etc.”

Answer 8: Here we would like to stick to the general use of "years" in publications.

Request 9: “Line 252 Table 1: I’d highly suggest the authors bracket the percentage to make your data presentation clearer.”

Answer 9: We thank the reviewer for this suggestion and have modified the presentation in Table 1 and Supplementary 1 accordingly.

Request 10: “Line 293 Figure 3, line 334 Figure 4: Please include a brief statistical description.”

Answer 10: As requested we have added a short statistical description in the figure legends.

Request 11: “Line 361 Figure 5: a) Please include the Figure 5 method in the methods section and make it clearer. b) Please include a clearer description of Figure 5 results and discussion.”

Answer 11: As requested we provide more information on the methodology in the methods section. Here we have adapted the corresponding paragraph as follows:

“Differential protein expression has been calculated using a moderated t-test as implemented in the R limma package (Version 3.5). Limma uses a test similar to the Student’s t-test in that it compares the means of NPX values for two groups of replicates for a given protein. The difference is in the calculation of variance. A Student’s t-Test calculates the variance from the data per protein. The moderated t-test uses information from all proteins to calculate variance. Differential pathway activity was assessed via a gene set enrichment analysis on the protein fold changes between EHMS and LHMS as implemented in the R gage package (Version 2.44) using the Gene Ontology Biological Processes as gene sets and p-value summarization via Stouffer's method. For the analysis only sets being represented by at least 3 proteins were considered. GSEA focuses on the differential expression of sets of related genes. It has advantages over per-gene based different expression analyses, including greater robustness, sensitivity, and biological relevance, as it calculates whether small, yet concerted changes in a set of biological entities, here proteins, shows significant regulation as a whole.”

Furthermore, we have extended the results section as follows: “In a final step, a gene set enrichment analysis was performed using the gene ontology biological processes as gene sets. Compared to the late metastasizing group, we found only the GO-terms “cell maturation” as significantly up-regulated in the serum of PDAC patients with EHMS (p-value=0.047). Contrary to this, factors that were elevated in the LHMS group could be associated with diverse processes: besides activation of MAPK activity or cellular responses to abiotic stimuli or estradiol factors could be associated particularly with migration of lymphocyte and mononuclear cells but also chemotaxis of natural killer cells (Figure 5). Overall, this analysis reveals a strong association of signif-icantly elevated factors in serum of LHMS patients with processes involved in recruitment and migration of immune cells.”

Request 12: “Table 1, Supplementary Table 2, Table 4, and other related parts: Did the authors mean clinical data only refers to “age at surgery and gender”? I’d highly suggest the authors consider rephrasing it to make it clearer.”

Answer 12: As requested we have modified the terms into “Patient demographics“ (for age, sex) and „pathological data“ for all pathological parameters.

Request 13: “In the current manuscript, LEGENDplexTM analysis (line 163 section 2.3.2) tests the protein level in the serum, and protein level biomarkers; the PCR-based Olink® analysis (line 175 section 2.3.3) shows gene expression level and transcriptional level biomarkers. The gene expression could not exactly predict its protein level. They are under different regulation mechanisms. It’s difficult to see what the advantages are that the authors listed two analyses high throughput versus other high throughput gene or protein analyses for clinical practice.”

Answer 13: We would like to point out that both methods detect the biomarkers on the protein level. However, Olink® analysis is based on another principle using antibody pairs labeled with complementary oligonucleotide strands which are then propagated via PCR. Hence, the Olink® technology does not evaluate gene expression but protein levels within the measured sample. The detailed procedure is described in chapter 2.3.3.

We very much hope that these modifications and additions sufficiently improved our paper making it now suitable for publication in Cancers.

Thank you very much for your consideration and efforts!

Sincerely yours,

S. Sebens, PhD

Round 2

Reviewer 2 Report

Thanks for the authors response, they addressed all my concerns about the manuscript.

I don't have any other quetions with this manuscript from my perspective. 

Author Response

We thank the reviewer for benevolent reply.

Reviewer 3 Report

Thank you for the revised manuscript.

1) Please make it clearer how you determined a set of data is “normally distribution” or “non-normally distribution” in the manuscript. This is very important.

2) Lines 137-138: Again, please confirm the exact “12 months” belong to early or late.

3) Lines 179-183 and throughout the manuscript: “1,1” should be “1.1” etc.

4) Line 180: “IL” is duplicated.

5) Line 300 Figure 2: Please italic p in the images.

Author Response

We thank the reviewer for the fast review and valuable comments. Accordingly, we have now prepared a further revised version of the manuscript. All changes have been highlighted by underlining.

Below, we will explain point-by-point how the arguments and criticisms of the reviewer have been dealt with.

Request 1: Please make it clearer how you determined a set of data is “normally distribution” or “non-normally distribution” in the manuscript. This is very important.

Answer 1: We have provided the following information in chapter 2.4: „Groups of datasets were tested for normal distribution and equal variance by Shapiro-Wilk and Equal Variance test, respectively.“

Request 2: Lines 137-138: Again, please confirm the exact “12 months” belong to early or late.

Answer 2: As requested we clearly indicate that 12 months belong to early hepatic spread and have this indicated in the modified manuscript as follows: ≤12 months, early hepatic metastatic spread (EHMS).

Request 3: Lines 179-183 and throughout the manuscript: “1,1” should be “1.1” etc.

Answer 3: As requested we have corrected the presentation of numbers.

Request 4: Line 180: “IL” is duplicated.

Answer 4: We have deleted the double IL.

Request 5: Line 300 Figure 2: Please italic p in the images

Answer 5: As requested we have corrected the presentation of the p-values in Figure 2A-C.

We very much hope that these modifications and additions sufficiently improved our paper making it now suitable for publication in Cancers.

Thank you very much for your consideration and efforts!

Sincerely yours,

S. Sebens, PhD

Round 3

Reviewer 3 Report

I checked the manuscript, the authors had made necessary modifications. I don't have further comments now. Just carefully check the clean version before publication.